# Primary Orbital Myxoid Liposarcoma

**DOI:** 10.3390/medsci11040072

**Published:** 2023-11-08

**Authors:** Miguel Armando Benavides-Huerto, Lourdes Páramo-Figueroa, Daniel Moreno-Páramo, Francisco Alejandro Lagunas-Rangel

**Affiliations:** 1Laboratory of Pathology and Cytopathology “Dr. Miguel Benavides”, Morelia 58260, Michoacán, Mexico; 2High Specialty Visual Clinic, Acámbaro 38600, Guanajuato, Mexico; 3Department of Genetics and Molecular Biology, Centro de Investigación y de Estudios Avanzados del Instituto Politécnico Nacional, Mexico City 07360, Mexico; 4Department of Surgical Sciences, Uppsala University, 752 36 Uppsala, Sweden

**Keywords:** cancer, adipocyte, eye, diagnosis

## Abstract

Although liposarcoma is the most prevalent soft tissue sarcoma in adults, head and neck liposarcomas are rare and account for less than 5% of all liposarcomas. The primary orbital location is even more exceptional, with fewer than 100 cases documented in the medical literature. Given the scarcity of cases of orbital liposarcoma and the limited familiarity of physicians and pathologists with this pathology, there is an increased risk of non-diagnosis or misdiagnosis, which may lead to inappropriate patient management. To address these challenges, we present a case of primary orbital myxoid liposarcoma and subsequently discuss the primary findings of this case based on the evidence documented in the medical literature. This comprehensive text is designed to serve as a valuable resource for healthcare professionals and pathologists, with the goal of promoting both clinical suspicion and accurate diagnosis and treatment of this rare condition in future cases.

## 1. Introduction

Liposarcomas are malignant tumors of adipocytic differentiation and represent one of the most prevalent subtypes of soft tissue sarcomas, constituting about 15% to 20% of all cases. This condition is categorically divided into four primary subtypes: well-differentiated liposarcoma (WDLPS, also known as atypical lipomatous tumor), dedifferentiated liposarcoma (DDLPS), myxoid liposarcoma (MLPS) and pleomorphic liposarcoma [1]. Although liposarcoma is most commonly seen in the muscles of the extremities or abdomen, it can also occur in the orbit, although this is rare [2,3,4,5,6,7,8,9]. The first case of orbital liposarcoma was documented by Strauss in 1911 [5,9], and to date (2023), the scientific literature has reported fewer than 100 documented cases of this rare condition. Interestingly, it is believed that the origin of the cells in ocular liposarcoma is not the mature adipocyte, but rather primitive stromal cells associated with intermuscular fascial planes or perivascular mesenchymal pluripotent mesenchymal cells [5,6,10]. Given the infrequency of cases of orbital liposarcoma and the limited exposure of physicians and pathologists to this condition, there is a clear need for increased documentation of these cases in the medical literature. With this aim in mind, we present a case of primary orbital myxoid liposarcoma and subsequently detail the primary findings of this case, along with those documented in the literature. This comprehensive text is intended to serve as an aid to health professionals and pathologists, both to encourage suspicion of this diagnosis and to facilitate its establishment and management in future cases.

## 2. Case Report

A 33-year-old male patient sought medical attention after experiencing two months of painless proptosis and swelling in the right upper eyelid (Figure 1A). At the time of presentation, magnetic resonance imaging was performed, which revealed the presence of a heterogenous mass located near the orbital roof (Figure 1B,C). An anterior orbitotomy was performed to remove the tumor, which measured approximately 6.4 × 2.5 × 3 cm and lacked a capsule (Figure 1D). After excision, the tumor was promptly submitted for histopathologic evaluation.

Histological analysis revealed a malignant mesenchymal neoplasm of adipose origin (Figure 2A). This neoplasm consisted of a mixture of adipose “signet ring” cells, some with stellate nuclei and others with a more spindle-shaped appearance (Figure 2B,C). Notably, these cells were positive for S-100 protein and vimentin and were embedded in a myxoid stroma (Figure 2D,E).

Furthermore, the lesion had a network of branching blood vessels and showed apparent CD-57 expression (Figure 2F).

Via reverse transcription-polymerase chain reaction (RT-PCR) (following previously established protocols [11,12]), we confirmed the presence of the fused in sarcoma (FUS)-DNA damage-inducible transcript 3 (DDIT3) fusion gene generated by the t(12;16) (q13;p11) translocation. In contrast, Ewing sarcoma breakpoint region 1 (EWSR1)-DDIT3 fusion gene generated by the t(12;22) (q13;q12) translocation was absent.

Based on these findings, the diagnosis of primary orbital myxoid liposarcoma was established. The initial medical recommendation for the patient was exenteration, but the patient chose to decline this procedure. However, given the presence of malignant cells at the margins of the initial surgical resection, a subsequent anterior orbitotomy became imperative to widen and ensure clear surgical margins. During this second intervention, the surgical margins were confirmed to be free of malignant cells. After postoperative recovery, the patient underwent adjuvant radiotherapy. The surgical site received a dose of 64 Gy by intensity-modulated radiation therapy (IMRT). At the last follow-up, performed three years after surgery, the patient is still alive with no signs of recurrence.

## 3. Discussion

Although liposarcoma is the most common soft tissue sarcoma in adulthood [1], it is exceptionally rare to occur in the orbit [9]. The first case is attributed to Strauss in 1911, and surprisingly, fewer than 100 cases have been documented since then [4,5]. According to a recent study conducted in the United States, orbital liposarcoma represents only 0.12% of all cases of liposarcoma in that country [9]. Interestingly, this particular type of tumor tends to manifest in individuals at an earlier age (most cases occurring between the third and sixth decade of life) compared to its counterparts located in the retroperitoneum, soft tissues or other areas of the body. It also shows a tendency to occur more frequently in women [5,9]. Although some cases have been reported suggesting a possible link between orbital liposarcoma and Li-Fraumeni syndrome (a genetic disorder that predisposes individuals to cancer due to germline mutations in the p53 tumor suppressor gene) [5,13], it is important to note that the available data are not yet conclusive to confirm this claim.

Orbital liposarcoma usually manifests as a painless proptosis [2,3,4,5,6,7,8], although in some cases, the presentation may include pain [14,15]. In addition, in some cases, the clinical presentation may be further complicated by the presence of symptoms such as epiphora (excessive tearing), diplopia (double vision) or decreased visual acuity [2,3,4,5,6,7,8]. It has been reported that the average duration of symptoms before diagnosis is usually about 5 months [5]. In the case we are presenting, the sole manifestation was painless proptosis, with no accompanying complications. From a radiological perspective, suspicion of orbital liposarcomas should arise when heterogeneous tumors presenting a combination of solid fatty and non-fatty components are found [3]. Obtaining a histopathologic diagnosis of orbital liposarcomas is complicated, often requiring diagnostic methods beyond conventional light microscopy and routine staining. Techniques such as immunohistochemistry and genetic testing are often employed to definitively confirm the diagnosis. In cases where there is initial histologic uncertainty, the use of immunohistochemical markers such as S-100 protein, vimentin, CD34, smooth muscle actin and desmin can significantly improve diagnostic accuracy [5]. Differential diagnosis includes inflammatory diseases (such as orbital pseudotumor), neurofibroma and myxoid pleomorphic lipoma, as well as other tumor possibilities, including lacrimal gland tumors and nonspecific orbital tumors [3,5]. In terms of subtype, myxoid liposarcoma emerges as the most prevalent histologic subtype in orbital liposarcomas, followed closely by the well-differentiated subtype [5,9]. A retrospective study conducted in a Chinese hospital yielded similar results [2].

Myxoid liposarcoma is distinguished by recurrent translocations, mainly t(12;16) (q13;p11) and, less frequently, t(12;22) (q13;q12), which result in fusion of FUS or EWSR1 with DDIT3, respectively [11]. However, data on the frequency of these fusion genes in orbital myxoid liposarcoma are currently unavailable. Both FUS-DDIT3 and EWSR1-DDIT3 fusion proteins are thought to function as aberrant transcription factors, although their mechanism of action is not known with certainty [16]. Among the mechanisms described, the FUS-DDIT3 fusion protein promotes proliferation and hinders adipogenic differentiation by upregulating Janus kinase (JAK)-signal transducer and activator of transcription (STAT) and insulin-like growth factor 1 receptor (IGF-IR)/phosphatidylinositol 3-kinase (PI3K)/AKT pathways [17,18]. It also represses peroxisome proliferator-activated receptor γ (PPARγ) and CCAAT/enhancer-binding protein α (C/EBPα) but activates eukaryotic translation initiation factor 4E (eIF4E) and C/EBPβ [19,20]. Regarding the EWSR1-DDIT3 fusion protein, its precise molecular mechanism of action has not been described to date (2023). It is worth emphasizing that it is imperative to increase molecular research in the context of orbital liposarcoma. Such studies are essential to advance our understanding and ability to identify the genetic, epigenetic and protein-related dysregulations that underlie this disease.

Because of the high recurrence rate and some resistance to radiotherapy and chemotherapy, surgery remains the primary approach to treat most orbital liposarcomas, with the possibility of removing adjacent tissues such as the eyelids, lateral crus and orbital wall to ensure complete eradication of the tumor [2,3,4,5,6,7,8]. However, the choice between globe-sparing local resections and initial exenteration depends on the surgeon’s preference and clinical judgment [9]. Complementary treatment options may also include radiotherapy (implantation of radioactive seeds or external focused radiotherapy) and, in selected cases, chemotherapy (with ifosfamide, vincristine and/or actinomycin) may also be considered, although it is used less frequently [2,3,4,5,6,7,8]. The efficacy of adjuvant radiotherapy in preventing the recurrence of liposarcomas remains a subject of debate and controversy in the medical community [3]. Recurrence is a frequent concern in this type of tumor. Unfortunately, local excision alone cannot guarantee nonrecurrence, so the more extensive procedure of exenteration of the eye is often preferred, despite being a difficult option for patients to accept [5]. In the present case, perhaps the ideal would have been to perform a biopsy to confirm the diagnosis before resorting to surgery for complete tumor removal. However, the surgical decision was motivated by a strong diagnostic suspicion and guided by the recommendations of the literature. The patient underwent two orbitotomies to completely remove the tumor and subsequently received radiotherapy. The prognosis has been good to date since, after an exhaustive 3-year follow-up, no recurrence has been detected. Although cases of recurrence after exenteration have been documented [10], early definitive surgery is likely to have had a curative outcome for these patients [5,6]. Some treatment options are currently under investigation. Due to the prevalence of MDM2 and CDK4 overexpression, significant research efforts have been devoted to therapeutically targeting these proteins. In addition, exportin 1 inhibitors (EXO1), tyrosine kinase inhibitors (TKI) and PPARγ agonists are being studied as potential treatment modalities [1]. Recently, immunotherapy has also shown promise as a treatment method for liposarcoma. This involves the use of immune checkpoint molecules such as programed death 1 (PD1), PD1 ligand (PDL1) and cytotoxic T-lymphocyte antigen 4 (CTLA4), as well as genetically modified T cells, encompassing T-cell receptors (TCRs) and chimeric antigen receptors (CARs) [21]. Notably, metastasis is a rare event in this type of tumor but there have been some reports mentioning them [5,22]. When assessing survival rates, it was observed that patients diagnosed with orbital liposarcoma face similar mortality rates as patients with soft-tissue liposarcoma, although they have lower mortality rates compared to patients with retroperitoneal liposarcoma [9]. Previous research has reported that the well-differentiated subtype has the most favorable prognosis [5]. This subtype is usually characterized by locally aggressive behavior, although it generally lacks metastatic potential unless dedifferentiation occurs. The dedifferentiation process is relatively infrequent, occurring in approximately 5% to 15% of cases, and this transformation usually takes an average of 7 to 8 years to manifest [9].

## 4. Conclusions

Orbital liposarcoma is an uncommon form of cancer that, due to its rarity, often receives limited attention from medical professionals and pathologists. This relative obscurity may increase the likelihood of misdiagnosis and consequently lead to inadequate clinical management affecting the patient’s prognosis. Besides conventional light microscopy and standard staining techniques, confirmation of the diagnosis usually requires immunohistological studies and, in some cases, more complex tests may be warranted. The main treatment option is usually exenteration of the eye, although this decision is understandably challenging for the patient. Furthermore, radiation therapy may be considered as part of the treatment plan. Recurrence is a common concern, underscoring the importance of close and vigilant patient follow-up.

## Figures and Tables

**Figure 1 medsci-11-00072-f001:**
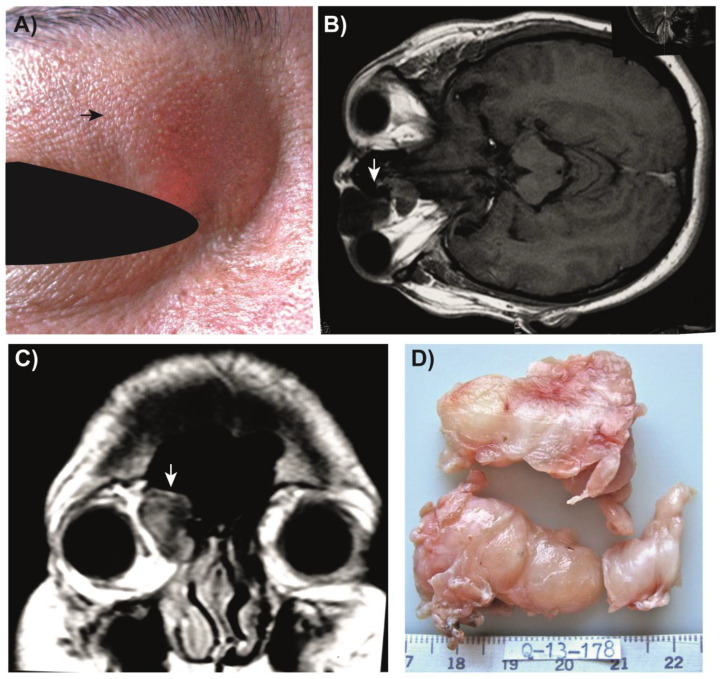
Clinical presentation of the case. (**A**) The patient’s most evident symptom was a painless proptosis (black arrow). (**B**) Axial magnetic resonance imaging (MRI) provided crucial diagnostic data (white arrow points to the tumor). (**C**) A frontal MRI clearly showed a tumor located near the orbital roof (white arrow). (**D**) Image of the tumor surgically removed during orbitotomy and subsequently subjected to histopathological analysis.

**Figure 2 medsci-11-00072-f002:**
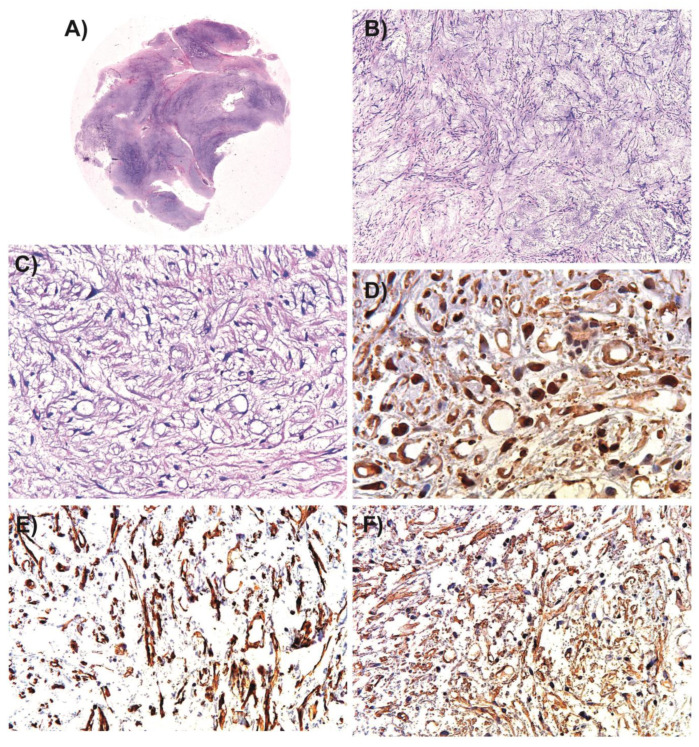
Histopathological analysis of the case. (**A**) Overview of the tumor neoplasm (H&E, 1×). (**B**) The neoplasm has a marked myxoid appearance with numerous branching blood vessels (H&E, 5×). (**C**) Signet ring cells and perineural invasion are evident at higher magnification (H&E, 10×). (**D**) Immunohistological analysis of S-100 protein (40×). (**E**) Immunohistological analysis of vimentin (40×). (**F**) Immunohistological analysis of CD-57 in tumor tissue (40×).

## Data Availability

Data sharing is not applicable to this article as no new data were created or analyzed in this study.

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
