# Peer review of "Primary Orbital Myxoid Liposarcoma"

_medsci, 2023, doi:10.3390/medsci11040072_

Round 1
Reviewer 1 Report
Comments and Suggestions for Authors
Dear editor,
Thank you for giving me an opportunity to review the manuscript that presents very rare orbital myxoid liposarcoma.
This case report seems meaningful, so I would like the authors to show the presence of fusion-gene like FUS/CHOP or DDIT3 because histological diagnosis is essential in this report.
Comments on the Quality of English Languageeasy to understand.
Author Response
Dear Reviewer
We thank you for taking the time to review our work and for your valuable comments, which help us to improve its quality. Responses to their comments are given below, and all modifications to the manuscript are indicated in red text.
Q1
This case report seems meaningful, so I would like the authors to show the presence of fusion-gene like FUS/CHOP or DDIT3 because histological diagnosis is essential in this report.
R1
This information was added to the manuscript and appears as follows:
Furthermore, by reverse transcription-polymerase chain reaction (RT-PCR) (following previously established protocols [11,12]), we confirmed the presence of the fused in sarcoma (FUS)-DNA damage-inducible transcript 3 (DDIT3) fusion gene generated by the t(12;16) (q13;p11) translocation. In contrast, Ewing sarcoma breakpoint region 1 (EWSR1)-DDIT3 fusion gene generated by the t(12;22)(q13;q12) translocation was absent.
Reviewer 2 Report
Comments and Suggestions for Authors
In my opinion, the manuscript of Benavides-Huerto et al. Primary orbital myxoid liposarcoma needs some clarifications.
It is not clear to me why in the first instance they decided on surgery with radical intent, which is rather difficult given the site, and did not opt for a diagnostic biopsy, postponing the choice between surgery and neoadjuvant therapy after histological examination. In case of a radio or chemo-sensitive tumor, it could have been a reasonable choice.
I think this could be explained, however briefly.
Second, I think the paper would be more complete with a description of technique, fields, and doses of Radiotherapy.
Author Response
Dear Reviewer
We thank you for taking the time to review our work and for your valuable comments, which help us to improve its quality. Responses to their comments are given below, and all modifications to the manuscript are indicated in red text.
Q1
In my opinion, the manuscript of Benavides-Huerto et al. Primary orbital myxoid liposarcoma needs some clarifications.
It is not clear to me why in the first instance they decided on surgery with radical intent, which is rather difficult given the site, and did not opt for a diagnostic biopsy, postponing the choice between surgery and neoadjuvant therapy after histological examination. In case of a radio or chemo-sensitive tumor, it could have been a reasonable choice.
I think this could be explained, however briefly.
R1
As you have rightly pointed out, the ideal would have been to perform a biopsy prior to radical surgery to confirm the diagnosis. However, the existing literature advises removal of the tumor and, in many cases, ocular exenteration because of the high recurrence rate associated with this tumor, even when it initially responds to radiation therapy. This was added to the manuscript and appears as follows:
In the present case, perhaps the ideal would have been to perform a biopsy to confirm the diagnosis before resorting to surgery for complete tumor removal. However, the surgical decision was motivated by a strong diagnostic suspicion and guided by the recommendation of the literature.Q2
Second, I think the paper would be more complete with a description of technique, fields, and doses of Radiotherapy.
R2
This information was incorporated and appears in the manuscript as follows:
After postoperative recovery, the patient underwent adjuvant radiotherapy. The surgical site received a dose of 64 Gy by intensity-modulated radiation therapy (IMRT).Round 2
Reviewer 1 Report
Comments and Suggestions for Authors
The authors addressed my suggested point.
This manuscript is acceptable for publication.